# The Role of Perfectionistic Self-Presentation in Pediatric Pain

**DOI:** 10.3390/ijerph18020591

**Published:** 2021-01-12

**Authors:** Elisabet Sánchez-Rodríguez, Alexandra Ferreira-Valente, Anupa Pathak, Ester Solé, Saurab Sharma, Mark P. Jensen, Jordi Miró

**Affiliations:** 1Universitat Rovira i Virgili, Unit for the Study and Treatment of Pain–ALGOS, Research Center for Behavior Assessment (CRAMC), Department of Psychology, 43007 Tarragona, Spain; elisabet.sanchez@urv.cat (E.S.-R.); ester.sole@urv.cat (E.S.); 2Institut d’Investigació Sanitària Pere Virgili, 43007 Tarragona, Spain; 3William James Center for Research, ISPA-Instituto Universitário, Rua Jardim do Tabaco No 34, 1149-041 Lisbon, Portugal; mafvalente@gmail.com; 4Department of Rehabilitation Medicine, University of Washington, Seattle, WA 98104, USA; mjensen@uw.edu; 5Centre for Musculoskeletal Outcomes Research, Department of Surgical Sciences, University of Otago, Dunedin 9054, New Zealand; anupa.pathak@postgrad.otago.ac.nz (A.P.); saurabsharma1@gmail.com (S.S.); 6Department of Physiotherapy, Kathmandu University School of Medical Sciences, Dhulikhel 45209, Nepal

**Keywords:** adolescents, pain intensity, pain interference, pain catastrophizing, perfectionistic self-presentation

## Abstract

This study sought to better understand the associations between perfectionistic self-presentation and measures of pain intensity, pain catastrophizing, pain interference, and fatigue in children and adolescents with pain. In the study, 218 adolescents responded to measures of perfectionistic self-presentation (i.e., perfectionistic self-promotion, nondisplay of imperfection and nondisclosure of imperfection), pain intensity, pain catastrophizing, pain interference, and fatigue. Four hierarchical regression analyses and three mediation analyses were conducted. Our results showed that perfectionistic self-promotion was significantly and independently associated with pain intensity and that nondisplay of imperfection was significantly and independently associated with pain catastrophizing, pain interference, and fatigue. Nondisclosure of imperfection was not significantly associated with any criterion variable. Pain catastrophizing mediated the association between both perfectionistic self-presentation and nondisplay imperfection and pain interference but not between nondisclosure of imperfection and pain interference. The findings provide new information about the role of perfectionistic self-presentation in children and adolescents’ experience of pain. These findings, if replicated, support perfectionism as a potential target of pain treatment in young people.

## 1. Introduction

Chronic pain is a widespread problem. Between 11% and 38% of children and adolescents worldwide have been reported to experience some form of chronic pain [1], and about 5% experience serious disability problems related to this pain [2]. In Spain, the prevalence of pediatric chronic pain is in line with these results. For example, Huguet and Miró [2] found that 37% of schoolchildren reported some form of chronic pain.

The biopsychosocial model of pain argues that a complex set of biological, psychological, and sociocultural variables contribute to the experience and impact of pain [3]. A psychological variable that is receiving a growing amount of empirical attention in the study of pediatric pain is perfectionism [4,5]. Hewitt and Flett’s Multidimensional Model of Perfectionism [6] identifies three domains of perfectionism. The first is trait perfectionism, defined as “…a personality disposition characterized by striving for flawlessness and setting exceedingly high standards of performance accompanied by overly critical evaluations of one’s behaviour” [7] (p. 171). It consists of three subdomains: Self-oriented perfectionism, other-oriented perfectionism, and socially prescribed perfectionism. The second domain is perfectionistic self-presentation, defined as an “interpersonal expression of perfection or the drive to appear to others as perfect by either publicly promoting one’s ‘perfection’ or by concealing one’s imperfections” [8] (p. 126). Perfectionistic self-presentation consists of the following three subdomains: Perfectionistic self-promotion, nondisplay of imperfection, and nondisclosure of imperfection. The third domain of Hewitt and Flett’s model consists of cognitive processes reflecting perfectionistic thoughts (e.g., ruminative thoughts regarding the need to be perfect). Figure 1 shows a schema of the Hewitt and Flett’s Multidimensional Model of Perfectionism.

Perfectionism has been shown to be positively associated with psychological inflexibility, that is, the inability to adapt our responses to the demands of the situation [9,10]. In fact, psychological inflexibility is considered as a core element of perfectionism, and has been shown to be related to avoidance behavior and psychological distress [11,12]. The associations between measures of trait perfectionism and psychological inflexibility with measures of function have been studied in a number of clinical populations and community samples. For example, trait perfectionism has been shown to be consistently related with measures of psychological dysfunction such as anxiety and depressive symptoms (including suicidal behaviors) in studies with adults [13,14] and children [15,16,17]. Perfectionistic self-presentation, although less studied in pediatric populations, has also shown to be significantly related to psychological distress. For example, Hewitt and colleagues [8] found that measures of the three domains of perfectionistic self-presentation were significantly associated with measures of depressive symptoms in adolescents with anxiety and depression disorders. They also found that their measure of nondisplay of imperfections was associated significantly with measures of depression, anxiety, and worry beyond the effects of perfectionism traits, and that nondisclosure of imperfection predicted depression, social anxiety, and anger beyond the effects of perfectionistic traits. Along these lines, Flett and colleagues [18] found that a measure of perfectionistic self-presentation in early adolescents (12 to 13 years old) from the community was significantly associated with social anxiety, even when controlling for measures of trait perfectionism.

Overall, then, both perfectionism as a trait and perfectionistic self-presentation have been consistently associated with a variety of psychological dysfunction domains (e.g., anxiety, depression, worry) [19], all of which have also been shown to play a role in the experience of pain (e.g., [20,21]). However, the study of perfectionism in samples of individuals with pain is very limited. The research that has been conducted suggests that perfectionism might play an important role in the experience of pain in both adult [14,22] and pediatric populations [4]. For example, Kempke and colleagues [23] found that measures of perfectionistic traits were significantly associated with the frequency and intensity of pain and fatigue in a sample of adult patients with chronic fatigue syndrome. In a subsequent study [22], these investigators found that self-critical perfectionism was significantly associated with poor treatment outcomes in a sample of adults with chronic pain.

Focusing in pediatric populations, Bonvanie and colleagues [24] found, in a longitudinal study carried out with adolescents, that trait perfectionism was associated with functional somatic symptoms (i.e., pain, headache, stomachache, nausea, vomiting, dizziness, and fatigue without an identified organic cause). Furthermore, a recent study by Randall and colleagues [5], carried out with children with chronic pain, showed that measures of trait perfectionism were significantly associated with physical disability through its relationship with the fear of pain and pain catastrophizing, and that parent’s perfectionism also played a role in their children’s pain experience.

Although these studies support the premise that perfectionism may play an important role in the adjustment to chronic pain, further studies are needed to determine the reliability and generalizability of these preliminary findings. Furthermore, to the best of our knowledge, no studies have yet explored the role that perfectionistic self-presentation could play in the experience of pain in children and adolescents.

Given these considerations, the primary aim of the current study was to better understand the relationships between perfectionistic self-presentation and measures of pain intensity, pain catastrophizing, pain interference, and fatigue in children and adolescents with pain. We hypothesized that the three domains of perfectionistic self-presentation (i.e., perfectionistic self-promotion, nondisplay of imperfection, and nondisclosure of imperfection) would evidence significant concurrent positive associations with measures of pain intensity, pain catastrophizing, pain interference, and fatigue. Furthermore, we sought to understand the potential moderating role of perfectionism in the associations between pain intensity and the other three pain-related criterion variables (i.e., pain catastrophizing, pain interference, and fatigue). We hypothesized that, if a significant moderating effect emerged, the association between pain intensity and the criterion variables would be stronger among children with higher levels of perfectionistic self-presentation than children with lower levels of perfectionistic self-presentation. Finally, we sought to determine if the mediating effects of pain catastrophizing on the association between perfectionism and pain interference found in previous studies [5] would be replicated in the current nonclinical sample of schoolchildren with pain. 

## 2. Materials and Methods

### 2.1. Participants

We invited children enrolled in grades 7 to 12 in 3 schools of the province of Tarragona (Catalonia, Spain) to participate in the study. We included schoolchildren who: (1) Were able to read and write Spanish; (2) were between 12 and 18 years old; (3) had experienced pain during the last 3 months, and (3) provided complete responses to all the questionnaires used in this study. Exclusion criteria were: (1) Failing to return the informed consent signed by their parents and (2) having any cognitive impairment that would interfere with the ability to understand and complete the questionnaires. 

A total of 500 adolescents from 12 to 18 years old were invited to participate in this study. Of these, 333 (67%) assented to participate and returned the informed consent signed by their parents and participated in the study. Of the participants, 26 (8%) reported not having experienced pain in the last 3 months and 89 (27%) did not complete all the measures and were excluded from final analyses. Thus, the final sample of the participants was made up of 218 adolescents (44% of the initially invited) that experienced pain in the 3 months preceding the study. The average age was 14.39 years old (SD = 1.79; range = 12 to 18 years). There were no statistically significant differences in the average age associated to sex. Table 1 provides additional descriptive information about the study sample.

### 2.2. Measures

Sociodemographic and pain information. Participants were asked to provide information about their age, sex, presence or absence of pain in the last 3 months, and pain location using a pain site checklist based on the site classification recommended by the International Association of Pain [25]. In addition, for each pain location, they were asked to rate the average intensity of that pain in the last 7 days using a 0–10 Numerical Rating Scale (NRS-11) where 0 represents “No pain” and 10 represents “Very much pain.” For participants who had multiple pain locations, we selected the highest average pain intensity rating (among those provided for each site) for data analyses. The NRS-11 has been shown to provide valid and reliable data in children as young as 6 years old [26,27]. 

Perfectionistic self-presentation. We assessed perfectionistic self-presentation using a Spanish version of the Perfectionistic Self-Presentation Scale–Junior Form (PSPS–Jr; [8]). With the PSPS–Jr, respondents are asked to rate the degree of agreement with each of the 18 statements included in the questionnaire using a 5-point Likert scale from 1 (“Strongly disagree”) to 5 (“Strongly agree”). The PSPS–Jr assesses 3 different domains of perfectionistic self-presentation: (1) Perfectionistic Self-Promotion (8 items, e.g., “I always have to look perfect”); (2) Nondisplay of Imperfection (P-ND; 6 items, e.g., “I feel bad about myself when I make mistakes in front of other people”), and (3) Nondisclosure of Imperfection (4 items, e.g., “I should always keep my problems secret”). The PSPS–Jr scales have shown to provide valid and reliable scores in several samples of children and adolescents [8,18]. Higher scores on each scale reflect a greater endorsement of each domain. In order to develop a translated version of the PSPS–Jr to use for this study, we conducted a back-translation process from English to Spanish following international guidelines [28]. First, 2 bilingual authors (ESR and JM) translated the original version into Spanish and agreed on a single translation. Next, a professional translator (a native English speaker and a linguist) back-translated the Spanish version into English. Then, the back-translated instructions and items were reviewed by the authors of the translated version (ESR and JM) with the professional translator to determine if any additional changes in the Spanish version were needed. At this stage, no changes were deemed necessary. The internal consistency (Cronbach’s alphas) of the 3 scales of the PSPS–Jr in our sample was excellent for the Perfectionistic Self-Promotion scale (α = 0.90) and borderline for Nondisplay of Imperfection and Nondisclosure of Imperfection scales (αs = 0.67 and 0.60, respectively).

Pain catastrophizing. Participants were asked to indicate the frequency with which they thought or felt 13 different catastrophizing responses to pain using the Spanish version of the Pain Catastrophizing Scale (PCS-C; [29,30]). Responses to each item can vary from 0 (“Not at all”) to 4 (“Always”). The PCS-C can be scored as a total score of pain catastrophizing or as 3 subscales assessing rumination, magnification, and helplessness. Higher scores on the PCS-C reflect higher levels of pain catastrophizing. In this study, we used the total score of the PCS-C. Reports from the PCS-C have been shown to be valid and reliable when used with children and adolescents [29,30]. The internal consistency (Cronbach’s alpha) of the PCS-C total scale score was good (*α* = 0.89) in our sample.

Pain interference. Pain interference was assessed using the 8-item pediatric PROMIS Pain Interference scale v.2.0 (PROMIS-PI; [31]). With this measure, respondents are asked to indicate the frequency with which pain has interfered with 8 different daily activities during the last 7 days using a 5-point Likert scale ranging from 1 (“Never”) to 5 (“Almost always”). Higher scores reflect higher pain interference. The pediatric version of the PROMIS-PI has been shown to provide a valid and reliable measure of pain interference in children and adolescents [32]. The internal consistency (Cronbach’s alpha) of the pediatric PROMIS-PI was good (α = 0.89) in our sample.

Fatigue. We used the Spanish version of the 4-item fatigue short form from the PROMIS Pediatric-25 Profile Form v.2. With these items, respondents are asked to rate how often they experience each fatigue response using a 5-point Likert scale from 1 (“Never”) to 5 (“Almost always”). Higher scores in the fatigue scale reflect greater fatigue. Previous work has shown that these PROMIS items are able to provide valid and reliable data for assessing fatigue in children and adolescents [33]. The internal consistency (Cronbach’s alpha) of the measure was good (α = 0.85) in our sample.

### 2.3. Procedure

We first contacted 4 secondary schools in the province of Tarragona (Catalonia, Spain) to inform them about the study, and 3 of them were willing to consider participation. We then met the principals of these 3 schools to describe the study procedures and goals and they consented for data collection. Next, we sent a letter to the parents of the children enrolled in grades 7 to 12 in these schools, explaining the study objectives and procedures and requesting permission for their children to participate in the study. Parents were requested to indicate their approval by signing an informed consent form and returning it to the researchers. Research staff then went to the participating schools and administered the questionnaires (self-reports) to the participants who had assented and whose parents provided consent. In order to foster truthfully responses, participants were told that there were no correct or incorrect answers and that all the answers would be anonymized. The study procedures were approved by the Ethics Committee of the *Institut d’Investigació Sanitària Pere i Virgili*. 

### 2.4. Data Analysis

We first computed descriptive statistics (means and standard deviations for continuous variables and number and percentages for dichotomous variables) for the demographic and study variables and computed zero-order correlation coefficients among the study variables for descriptive purposes. Next, we examined the distributions (skewness and kurtosis) and the multicollinearity (by computing variance inflation factors and tolerance) of the predictor and criteria variables to ensure that they met the assumptions for the planned analyses (i.e., normal distribution and no multicollinearity). Then, in order to test the first and second hypotheses (i.e., that the 3 domains of perfectionistic self-presentation would evidence significant concurrent positive association with measures of pain intensity, pain catastrophizing, pain interference, and fatigue and that perfectionistic self-presentation could act as moderator in the associations between pain intensity and the other three pain-related criterion variables), we computed 4 hierarchical regression analyses, with the measures of pain intensity, pain interference, pain catastrophizing, and fatigue as the criterion variables. We entered demographic variables (sex and age) in step 1 as control variables. In step 2, we entered pain intensity (only when the criterion variable was pain catastrophizing, pain interference, or fatigue). In step 3, we entered the 3 domains of perfectionistic self-presentation (i.e., perfectionistic self-promotion, nondisplay of imperfection, and nondisclosure of imperfection). Finally, in step 4, we entered the 3 interaction terms (Pain Intensity x Perfectionistic Self-Promotion, Pain Intensity x Nondisplay of Imperfection, and Pain Intensity x Nondisclosure of Imperfection) when the criterion variables were pain catastrophizing, pain interference, or fatigue. If an interaction term emerged as significant, we planned to evaluate the moderating effects using the visualization strategy recommended by Hayes and Rockwood using PROCESS macro for SPSS [34].

Finally, in order to evaluate the potential mediating role of pain catastrophizing in the association between the 3 domains of perfectionistic self-presentation and pain interference, we performed 3 mediation analyses using PROCESS macro, with the 3 domains of perfectionistic self-presentation as predictors; pain interference as the criterion; pain catastrophizing as potential mediators; and sex, age, and pain intensity as covariates. 

All data analyses were performed using the Statistical Package for Social Sciences (Windows version 25.0, SPSS Inc., Chicago, IL, USA). The sample size needed in the regression analyses was calculated using G*Power software v.3.1.9.4. (HHU, Heinrich Heine Universität Düsseldorf, Düsseldorf, North Rhine-Westphalia, Germany) Assuming 9 predictors and an alpha of 0.05, we needed a minimum of 166 participants to detect a medium effect (f^2^ = 0.15). 

## 3. Results

### 3.1. Description of the Study Sample

Table 2 shows descriptive statistics for the study variables. All the variables were normally distributed (skewness = −0.57 to 0.27 and kurtosis = −0.60 to −0.07) and multicollinearity was not a problem in our sample (all variance inflation factors (VIF) were lower than 10). Table 3 shows the Pearson correlations between the criteria and the predictor variables.

### 3.2. Regression Analysis Predicting Pain Intensity

The results of the regression analysis predicting pain intensity are presented in Table 4. As can be seen, age and sex explained a significant 4% of the variance of pain intensity in step 1. However, neither of them explained a statistically significant amount of the variance in pain intensity when the other was controlled for. After controlling for sociodemographic data, perfectionistic self-presentation as a whole explained an additional 4% of the variance in pain intensity (*p* = 0.042) in step 2. However, only perfectionist self-promotion (*β*= −0.23, *p* < 0.01) made a statistically significant and independent contribution to the prediction of pain intensity.

### 3.3. Regression Analysis Predicting Pain Catastrophizing

The results of the regression analysis predicting pain catastrophizing are presented in Table 5. As can be seen, age and sex explained a significant 8% of the variance of pain catastrophizing in step 1, due mainly to the effect of sex (higher levels of pain catastrophizing were reported by female participants; *β* = 0.28, *p* < 0.001). However, pain intensity did not explain a significant amount of the variance in pain catastrophizing in step 2 (*R*^2^ change = 0.02, *p* > 0.05). In step 3, after controlling for sociodemographic variables and pain intensity, perfectionistic self-presentation explained an additional 15% of the variance of pain catastrophizing, due mainly to the effects of nondisplay of imperfection (*β* = 0.40, *p* < 0.001). None of the interaction terms emerged as significant in step 4 (*R*^2^ change = 0.02, *p* > 0.05).

### 3.4. Regression Analysis Predicting Pain Interference

The results of the regression analysis predicting pain interference are presented in Table 6. As can be seen, age and sex explained a significant 3% of the variance of pain interference in step 1, due mainly to the effect of sex (higher levels of pain interference were reported by female participants; *β* = 0.14, *p* < 0.05), and pain intensity explained an additional 11% of its variance in step 2 (*β* = 0.32, *p* < 0.001). After controlling for the sociodemographic variables and pain intensity, perfectionistic self-presentation explained an additional significant 6% of the variance of pain interference, due mainly to the effects of nondisplay of imperfection (*β* = 0.22, *p* < 0.05). None of the interaction terms emerged as significant in step 4 (*R*^2^ change= 0.02, *p* > 0.05). 

### 3.5. Regression Analysis Predicting Fatigue

The results of the regression analysis predicting fatigue are presented in Table 7. As can be seen, age and sex explained a significant 7% of the variance of fatigue in step 1, due mainly to the effect of sex (higher levels of fatigue were reported by female participants; *β* = 0.25, *p* < 0.01), and pain intensity explained an additional 4% of the variance (*β* = 0.18, *p* < 0.01) (*β* = 0.18, *p* < 0.01) in step 2. After controlling for the sociodemographic variables and pain intensity, perfectionistic self-presentation explained an additional significant 5% of the variance in fatigue, due mainly to the effects of nondisplay of imperfection (*β =* 0.24, *p* < 0.01). None of the interaction terms emerged as significant in step 4 (*R*^2^ change= 0.01, *p* > 0.05).

### 3.6. Mediation Analysis

In order to evaluate the mediation effects of pain catastrophizing on the association between perfectionistic self-presentation and pain interference, we conducted three different mediation analyses (one for each Perfectionistic Self-Presentation scale). In the first model (see Figure 2), we tested the mediation effects of pain catastrophizing on the association between perfectionistic self-promotion (predictor) and pain interference (criterion). We also added sex and pain intensity as covariates. In support of this model, perfectionistic self-promotion was found to be significantly associated with pain catastrophizing (path a: *β* = 0.31, *p* < 0.01). In addition, pain catastrophizing was significantly associated with pain interference (path b: *β* = 0.39, *p* < 0.001). Furthermore, the direct effect of perfectionistic self-promotion on pain interference (path c’: *β* = 0.02 *p* = 0.77) was substantially less when catastrophizing was included in the model than when it was not included in the model (path c: *β* = 0.14, *p* = 0.08) when controlling for pain catastrophizing. The reliability of the indirect effect was tested using the Bootstrapping method, and the statistical significance of the mediating role of pain catastrophizing *(β* = 0.12, 95% confidence interval (CI) = 0.0476 to 0.2052) was confirmed. Finally, with respect to the covariate variables (i.e., sex and pain intensity), sex was significantly associated with the mediator (i.e., pain catastrophizing), and pain intensity was significantly associated with both the mediator and the criterion variable (i.e., pain interference). 

In the second model (see Figure 3), we tested the mediation effects of pain catastrophizing on the association between nondisplay of imperfection (predictor) and pain interference (criterion). We also added the variable sex and pain intensity as covariates. In support of this model, nondisplay of imperfection was found to be significantly associated with pain catastrophizing (path a: *β* = 0.79, *p* < 0.001). In addition, pain catastrophizing was significantly associated with pain interference (path b: *β* = 0.36, *p* < 0.001). Furthermore, although the total effect of nondisplay of imperfection on pain interference was significant (path c: *β* = 0.43 *p* < 0.001), this effect became nonsignificant (path c’: *β* = 0.15 *p* = 0.239) when controlling for pain catastrophizing. The significance of the indirect effect was tested using the Bootstrapping method and the statistical significance of the mediating role of pain catastrophizing (*β* = 0.29, 95% CI = 0.1622 to 0.4319) was confirmed. With respect to the covariate variables (i.e., sex and pain intensity), sex was significantly associated with the mediator (i.e., pain catastrophizing), and pain intensity was significantly associated with the criterion variable (i.e., pain interference). 

Finally, in the third model (see Figure 4), we tested the mediation effects of pain catastrophizing on the association between nondisclosure of imperfection (predictor) and pain interference (criterion). We also added the variable sex and pain intensity as covariates. We found that nondisclosure of imperfection was significantly associated with pain catastrophizing (path a: *β* = 0.41, *p* < 0.05). In addition, pain catastrophizing was significantly associated with pain interference (path b: *β* = 0.38, *p* < 0.001). Furthermore, while the total effect of nondisclosure of imperfection on pain interference was significant (path c: *β* = 0.38, *p* < 0.05), this effect became nonsignificant (path c’: *β* = 0.23, *p* = 0.150) when controlling for pain catastrophizing. However, the mediating role of pain catastrophizing on the association between nondisclosure of imperfection and pain interference could not be confirmed because the indirect effect was not statistically significant (*β* = 0.15, 95% CI = −0.0005 to 0.0344). With respect to the covariate variables (i.e., sex and pain intensity), sex was significantly associated with the mediator (i.e., pain catastrophizing), and pain intensity was significantly associated with the criterion variable (i.e., pain interference).

## 4. Discussion

The primary objective of this study was to better understand the role that perfectionistic self-presentation plays in explaining the pain experience of children and adolescents with pain recruited from the community. We hypothesized that the three domains of perfectionistic self-presentation would evidence significant concurrent positive associations with measures of pain intensity, pain catastrophizing, pain interference, and fatigue in this sample. We also hypothesized that, if a significant moderating effect of perfectionism was found, then the association between pain intensity and the criterion variables would be stronger among children with higher levels of perfectionistic self-presentation than children with lower levels of perfectionistic self-presentation. Finally, we sought to determine if the mediating effect of pain catastrophizing on the association between perfectionistic self-presentation and pain interference found in prior research would be replicated. 

The study hypothesis regarding the prediction of pain-related criterion variables from measures of perfectionism was partially supported. Some, but not all, of the domains of perfectionism showed significant associations with some criterion measures (see Figure 5). 

This result is consistent with previous research indicating that different domains of perfectionistic self-presentation might be distinctly associated with different outcomes [8,18]. Specifically, the findings suggest that nondisplay of imperfection could play a larger role in the adolescents’ experience of pain than either perfectionistic self-promotion or nondisclosure of imperfection. If this finding is replicated in future studies, including longitudinal studies supporting a causal influence of nondisplay of imperfection on pain, this would support the potential importance of targeting this component of perfectionism in pain management programs. Such programs might include Acceptance and Commitment Therapy to help develop or reinforce psychological flexibility (i.e., acceptance, cognitive defusion, contact with the present moment, self-as-context, committed action, values orientation) [35], or cognitive-behavioral therapy [36], with the objective of encouraging adolescents to participate in activities with which they might not evidence perfection while, at the same time, practicing self-acceptance. Research to evaluate the potential benefits of such programs, as well as the potential mediation role of reductions in nondisplay of imperfection in those benefits, is warranted.

The findings did not support a moderating role for perfectionistic self-presentation in the association between pain intensity and pain catastrophizing, pain interference, and fatigue. Thus, higher levels in perfectionistic self-presentation domains do not appear to make adolescents more vulnerable to the negative effects of pain intensity on function (see Figure 6). 

This is, to the best of our knowledge, the first study that tested the potential moderating role of perfectionistic self-presentation in the associations between pain intensity and the three pain-related criterion variables studied here (i.e., pain catastrophizing, pain interference, and fatigue). If these findings are replicated in other samples, they would suggest that teaching adolescents skills to reduce perfectionistic self-presentation, while possibly having some positive effects on some outcomes (e.g., increasing self-esteem, decreasing fear and intolerance of uncertainty; [37]), would not necessarily buffer the negative effects of pain on function. 

The study findings support the mediating effects of pain catastrophizing on the association between perfectionistic self-presentation and pain interference (see Figure 1 and Figure 2), although pain catastrophizing did not mediate the association between nondisclosure of imperfection and pain interference (see Figure 3). Previous work with pediatric pain populations [5] found that trait perfectionism (i.e., self-oriented, socially prescribed, and effortless perfectionism) impacts functional disability through its influence on pain catastrophizing. Our findings, in light of those reported by Randall and colleagues, suggest that trying to appear to others as perfect may contribute to increases in rumination and worry about pain, which could then lead to greater pain interference. If this finding is supported by future longitudinal research in this area, treatments which reduce perfectionism could potentially also decrease catastrophizing in adolescents, which could ultimately increase their function.

The study findings also suggest that not only a perceived *need* to be perfect (trait perfectionism) but also that a perceived need to *appear* perfect to the others (perfectionistic self-presentation) have an important role in how young people experience and cope with pain. As this is the first study to evaluate the role of perfectionistic self-presentation in a sample of schoolchildren with pain, additional research, especially longitudinal studies, is needed to determine which of the findings reported here could be replicated in future studies. 

This study has a number of limitations that should be considered when interpreting the results. First, our sample was a convenience community sample of schoolchildren with pain who may or may not be representative of the population of young people with pain. Future research should study perfectionism on additional samples of children and adolescents with pain to determine the generalizability of the findings. Second, because this was a cross-sectional study, we were not able to draw causal conclusions regarding the influence of perfectionistic self-presentation on pain and its impact. Longitudinal studies are also needed to understand causal associations between perfectionistic self-presentation and pain experience in pediatric samples. Third, all the data were collected via self-report. As a result, some of the significant associations found could have been due to shared method bias [38]. Future researchers should evaluate some of the domains using alternative strategies when possible (e.g., measuring function via an actigraph or using observational measures, or via parent-proxy reports) to ascertain whether the findings in this study are due to shared method bias. Fourth, the Spanish version of the PSPS–Jr used in this study had borderline levels of internal consistency for the Nondisplay of Imperfection and the Nondisclosure of Imperfection scale (αs = 0.67 and 0.60, respectively). Whereas these levels of reliability are consistent with those obtained with the original version of the questionnaire [8,18], it is possible that the limited reliability of these measures resulted in an underestimation of the true associations among the study variables. Given the potential relevance that trait perfectionism and perfectionistic self-presentation may have in the adjustment to pain among children and adolescents, it would be reasonable to consider revising the available measures (e.g., by adding new items) or developing more reliable measures of these domains in order to increase the reliability of the research findings.

## 5. Conclusions

The findings provide important new key information about the role of perfectionistic self-presentation in the children and adolescents’ experience of pain. To our knowledge, this is the first study that shows how adolescents need to appear as perfect to others is related to higher levels of pain severity. These findings, if replicated, would indicate the suitability of including perfectionism as a treatment objective in preventive programs for pain in young people.

## Figures and Tables

**Figure 1 ijerph-18-00591-f001:**
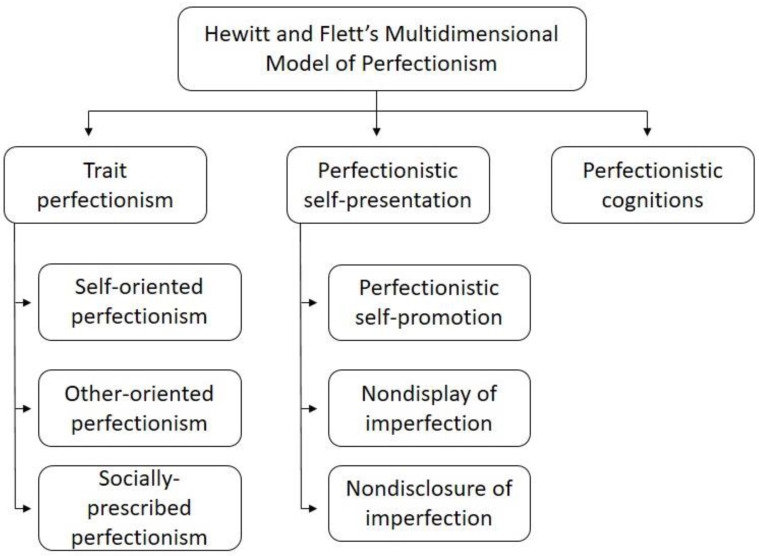
Hewitt and Flett’s Multidimensional Model of Perfectionism.

**Figure 2 ijerph-18-00591-f002:**
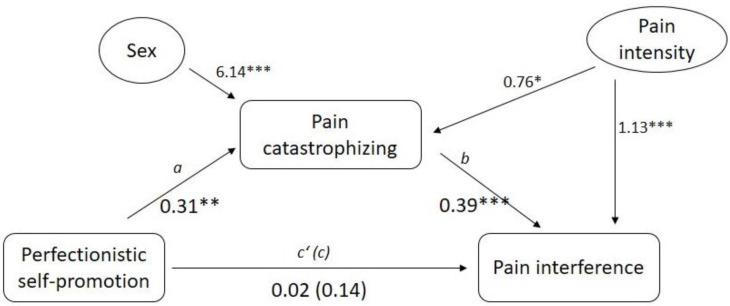
Mediation effects of pain catastrophizing on the association between perfectionistic self-promotion and pain interference. Note: * *p* < 0.05, ** *p* < 0.01, *** *p* < 0.001.

**Figure 3 ijerph-18-00591-f003:**
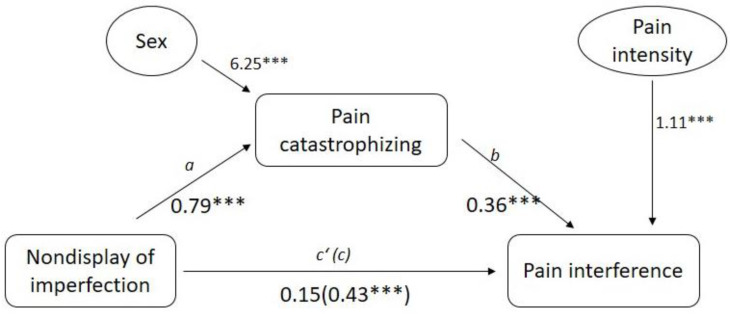
Mediation effect of pain catastrophizing on the association between nondisplay of imperfection and pain interference. Note: *** *p* < 0.001.

**Figure 4 ijerph-18-00591-f004:**
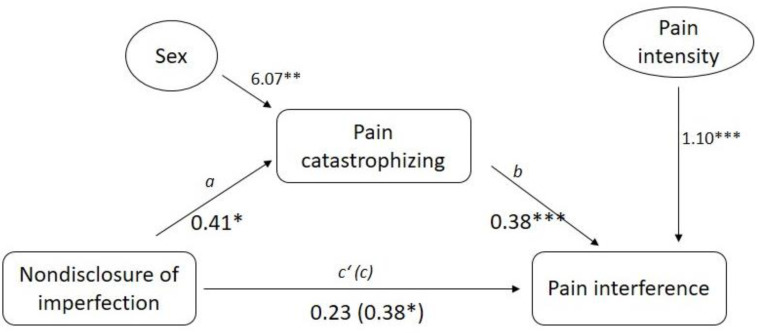
Mediation effect of pain catastrophizing on the association between nondisclosure of imperfection and pain interference. Note: * *p* < 0.05, ** *p* < 0.01, *** *p* < 0.001.

**Figure 5 ijerph-18-00591-f005:**
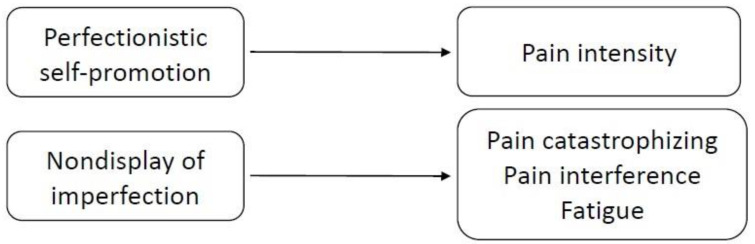
Predictions of pain-related criterion variables from measures of perfectionistic self-presentation.

**Figure 6 ijerph-18-00591-f006:**
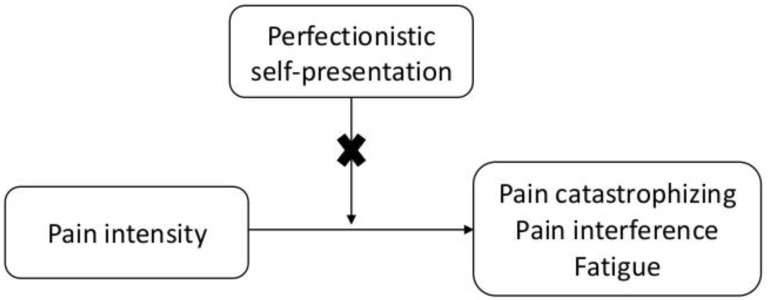
Moderating role for perfectionistic self-presentation in the association between pain intensity and pain catastrophizing, pain interference, and fatigue.

**Table 1 ijerph-18-00591-t001:** Descriptive data for the study sample (N = 218).

	Participants N (%)		
	Total (N = 218)	Female (N = 135)	Male (N = 83)
**Age**			
12	41 (19)	26 (19)	15 (18)
13	48 (22)	30 (22)	18 (22)
14	27 (12)	15 (11)	12 (14)
15	25 (12)	18 (14)	7 (8)
16	45 (21)	23 (17)	22 (27)
17	29 (13)	21 (16)	8 (10)
18	3 (1)	2 (1)	1 (1)
**Pain location**			
Head	160 (73)	112 (83)	48 (58)
Back	124 (57)	86 (64)	38 (46)
Belly/pelvis	124 (57)	100 (74)	24 (29)
Legs	95 (44)	56 (41)	39 (47)
Neck	84 (39)	49 (36)	35 (42)
Feet	68 (31)	35 (26)	33 (40)
Shoulders	62 (28)	39 (29)	23 (28)
Arms	43 (20)	23 (17)	20 (24)
Chest/breast	42 (19)	29 (21)	13 (16)
Hands	38 (17)	23 (17)	15 (18)
Bottom/hips	31 (14)	22 (16)	9 (11)
Other locations	17 (8)	11 (8)	6 (7)

**Table 2 ijerph-18-00591-t002:** Descriptive statistics for the study variables.

Domain	Mean (SD)	Range	Skewness	Kurtosis
Pain intensity (NRS-11)	5.62 (2.27)	0–10	−0.57	−0.07
Perfectionistic self-promotion (PSPS–J)	21.50 (8.21)	8–40	0.27	−0.64
Nondisplay of imperfection (PSPS–J)	17.97 (5.04)	6–29	0.06	−0.54
Nondisclosure of imperfection (PSPS–J)	11.18 (3.65)	4–20	0.15	−0.44
Pain Catastrophizing (PCS-C)	20.56 (10.90)	0–50	0.21	−0.58
Pain Interference (PROMIS-PI-8a)	52.42 (9.37)	34–78	0.19	−0.10
Fatigue (PROMIS-P-25)	52.25 (10.99)	35.4–77.6	0.22	−0.60

Note: NRS-11 = 0–10 Numerical Rating Scale; PSPS–J = Perfectionistic Self-Presentation Scale–Junior; PCS-C = Pain Catastrophizing Scale for Children; PROMIS-PI-8a = Eight-item pediatric PROMIS Pain Interference scale; PROMIS-P-25 = Four-item fatigue short-form scale from the PROMIS Pediatric-25 Profile Form v.2.

**Table 3 ijerph-18-00591-t003:** Pearson correlations between measures of perfectionistic self-presentation and the study criterion variables.

Criterion Variables (Pain and Function)	Perfectionistic Self-Presentation Variables (PSPS–J)		
	Perfectionistic Self-Promotion	Nondisplay of Imperfection	Nondisclosure of Imperfection
Pain intensity (NRS-11)	−0.17 *	0.06	0.05
Pain Catastrophizing (PCS-C)	0.19 **	0.36 ***	0.13 *
Pain Interference (PROMIS-PI-8a)	0.04	0.23 ***	0.18 **
Fatigue (PROMIS-P-25)	−0.03	0.21 **	0.15 *

* *p* < 0.05; ** *p* < 0.01; *** *p* < 0.001.

**Table 4 ijerph-18-00591-t004:** Regression analysis predicting pain intensity (NRS-11).

Step	Predictor	*R* ^2^	*R*^2^ Change	*F*	*β*	*t*	*p*	Tolerance	VIF
1	**Demographic variables**	0.04	0.04	3.40			**0.036**		
	Age				0.11	1.43	0.156	0.956	1.05
	Sex				0.13	1.80	0.073	0.997	1.00
2	**Perfectionistic self-presentation (PSPS–J)**	0.08	0.04	2.80			**0.042**		
	Perfectionistic Self-Promotion				−0.23	2.81	**0.005**	0.777	1.29
	Nondisplay of Imperfection				0.14	1.55	0.122	0.604	1.66
	Nondisclosure of Imperfection				0.03	0.30	0.764	0.686	1.46

Note: *p* values in boldface are significant at <0.05.

**Table 5 ijerph-18-00591-t005:** Regression analysis explaining pain catastrophizing (PCS-C).

Step	Predictor	*R* ^2^	*R*^2^ Change	*F*	*β*	*t*	*p*	Tolerance	VIF
1	**Demographics**	0.08	0.08	8.25			**<0.001**		
	Age				0.01	0.11	0.912	0.945	1.06
	Sex				0.28	4.24	**<0.001**	0.979	1.02
2	**Pain intensity (NRS-11)**	0.10	0.02	2.95	0.12	1.75	0.088	0.920	1.09
3	**Perfectionistic self-presentation (PSPS–J)**	0.23	0.15	11.86			**<0.001**		
	Perfectionistic Self-Promotion				0.08	0.99	0.322	0.744	1.38
	Nondisplay of Imperfection				0.40	4.68	**<0.001**	0.596	1.68
	Nondisclosure of Imperfection				−0.10	1.26	0.211	0.686	1.46
4	Interaction terms	0.27	0.02	1.67			0.174		

Note: *p* values in boldface are significant at <0.001.

**Table 6 ijerph-18-00591-t006:** Regression analysis explaining pain interference (PROMIS-PI-8a).

Step	Predictor	*R* ^2^	*R*^2^ Change	*F*	*β*	*t*	*p*	Tolerance	VIF
1	**Demographics**	0.03	0.03	3.18			0.044		
	Age				−0.07	0.93	0.353	0.945	1.06
	Sex				0.14	2.12	0.035	0.979	1.02
2	**Pain intensity (NRS-11)**	0.14	0.11	22.67	0.32	4.61	<0.001	0.920	1.09
3	**Perfectionistic self-presentation (PSPS–J)**	0.20	0.06	4.38			0.005		
	Perfectionistic Self-Promotion				0.01	0.07	0.944	0.744	1.35
	Nondisplay of Imperfection				0.22	2.51	0.013	0.596	1.68
	Nondisclosure of Imperfection				0.04	0.51	0.612	0.686	1.46
4	Interaction terms	0.22	0.02	1.30			0.276		

Note: *p* values in boldface are significant at <0.05.

**Table 7 ijerph-18-00591-t007:** Regression analysis explaining fatigue (PROMIS-P-25).

Step	Predictor	*R* ^2^	*R*^2^ Change	*F*	*β*	*t*	*p*	Tolerance	VIF
1	**Demographics**	0.07	0.07	6.39			**0.002**		
	Age				0.04	0.54	0.588	0.945	1.06
	Sex				0.22	3.09	**0.002**	0.979	1.02
2	**Pain intensity (NRS-11)**	0.11	0.04	8.45	0.18	2.55	**0.012**	0.920	1.09
3	**Perfectionistic self-presentation (PSPS–J)**	0.16	0.05	3.63			**0.014**		
	Perfectionistic Self-Promotion				−0.08	0.93	0.353	0.744	1.35
	Nondisplay of Imperfection				0.24	2.71	**0.007**	0.596	1.68
	Nondisclosure of Imperfection				0.03	0.30	0.768	0.686	1.46
4	**Interaction terms**	0.17	0.01	0.53			0.664		

Note: *p* values in boldface are significant at <0.05.

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
