# Peer review of "The Role of Perfectionistic Self-Presentation in Pediatric Pain"

_ijerph, 2021, doi:10.3390/ijerph18020591_

Round 1

Reviewer 1 Report

Review

Thank you for the opportunity to read this article. The topic taken up by the authors is very interesting. Increasingly, young people complain about the occurrence of pain and the inability to cope with it. The work is very interesting, especially that the research presented in it was conducted on healthy youth.

Comments:

Lines 97-111 - the aim of the work is very extensive with two independent hypotheses "0" - the authors should focus on one hypothesis - then the work would be clearer and easier to understand by a wider group of recipients

MATERIALS AND METHODS

Participants - no information about the group size - later this information appears in the results part, and it is a description of the group. There is no information about the average age of the respondents, broken down by gender. In the results, the authors observed statistically significant relationships between the sex and the examined aspects of pain, therefore the study group was not divided according to gender. Such a division is very important because the pain is felt differently by women and men, in addition, in the study group as many as 124 people indicated belly / pelvis pain was it related to menstruation, or also in men?

Line 115-120 - lack of information as to whether the health condition (a disease associated with the appearance of pain) was an element that excludes children from the study, who are chronically ill on drugs, but still able to learn.

There is no information on the use of painkillers - adolescents aged 15-18 years often use them.

General thoughts

  In its present form, with two research hypotheses developed in such detail, the work is addressed to a narrow group of specialists. The authors could focus their work on one hypothesis, use a gender division, supplement the dependencies on the use of OTC - over the counter preparations, and then the article would be available to a wider audience, not only for doctors but also psychologists, educators, physiotherapists and teachers.

Reviewer 2 Report

This study provides a broad insight into perceptionistic self presentation for Spanish teenagers. In detail, this study presents the associations between perfectionistic self presentation and measures of pain intensity, pain catastrophizing, pain interference, and fatigue in pediatric pain. It is also judged to be a high-quality paper by providing various analyses with rich data set.

However, to improve the overall quality of the paper, the following items need to be modified or supplemented.

-In the introduction, too abstract and lengthy sentences are described. Key concepts and definitions should be summarized in chart or picture form.

-In Materials and Methods, explain how you can prove that teenagers answered the questionnaire truthfully.

-How can you justify that the Perfectionistic Self-Presentation Scale – Junior Form(as well as PCS-C, PROMIS-PI, and so on)

 is an objective and fair indicator for this research?

-How can you objectify and generalize the survey results in a specific area(the province of Tarragona)?

-In Discussion, use tables or pictures instead of a lengthy sentence expression.

-change title from 'Conclusions' to 'Conclusions and Future Research Works", present some realistic future research issues.
